# Characteristics of Ventricular Electrophysiological Substrates in Metabolic Mice Treated with Empagliflozin

**DOI:** 10.3390/ijms22116105

**Published:** 2021-06-05

**Authors:** Shih-Jie Jhuo, I-Hsin Liu, Wei-Chung Tasi, Te-Wu Chou, Yi-Hsiung Lin, Bin-Nan Wu, Kun-Tai Lee, Wen-Ter Lai

**Affiliations:** 1Division of Cardiology, Department of Internal Medicine, Kaohsiung Medical University Hospital, Kaohsiung 80701, Taiwan; jhuoshihjie@gmail.com (S.-J.J.); ihsin343@yahoo.com.tw (I.-H.L.); azygo91@gmail.com (W.-C.T.); tewuchou@gmail.com (T.-W.C.); caminolin@gmail.com (Y.-H.L.); wtlai@cc.kmu.edu.tw (W.-T.L.); 2Graduate Institute of Clinical Medicine, Kaohsiung Medical University, Kaohsiung 80701, Taiwan; binnan@kmu.edu.tw; 3Department of Internal Medicine, Faculty of Medicine, College of Medicine, Kaohsiung Medical University, Kaohsiung 80701, Taiwan; 4Regeneration Medicine and Cell Therapy Research Center, Kaohsiung Medical University, Kaohsiung 80701, Taiwan

**Keywords:** SGLT2, empagliflozin, antiarrhythmic, cardiac remodeling, fibrosis, connexin

## Abstract

Empagliflozin (EMPA) is a sodium–glucose transporter 2 (SGLT2) inhibitor that functions as a new-generation glucose-lowering agent and has been proven to be beneficial for patients with cardiovascular diseases. However, the possible benefits and mechanisms of its antiarrhythmic effects in cardiac tissue have not yet been reported. In this study, we elucidated the possible antiarrhythmic effects and mechanisms of EMPA treatment in cardiac tissues of metabolic syndrome (MS) mice. A total of 20 C57BL/6J mice (age: 8 weeks) were divided into four groups: (1) control group, mice fed a standard chow for 16 weeks; (2) MS group, mice fed a high-fat diet for 16 weeks; (3) EMPA group, mice fed a high-fat diet for 12 weeks and administered EMPA at 10 mg/kg daily for the following 4 weeks; and (4) glibenclamide (GLI) group, mice fed a high-fat diet for 12 weeks and administered GLI at 0.6 mg/kg daily for the following 4 weeks. All mice were sacrificed after 16 weeks of feeding. The parameters of electrocardiography (ECG), echocardiography, and the effective refractory period (ERP) of the left ventricle were recorded. The histological characteristics of cardiac tissue, including connexin (Cx) expression and fibrotic areas, were also evaluated. Compared with the MS group, the ECG QT interval in the EMPA group was significantly shorter (57.06 ± 3.43 ms vs. 50.00 ± 2.62 ms, *p* = 0.011). The ERP of the left ventricle was also significantly shorter in the EMPA group than that in the GLI group (20.00 ± 10.00 ms vs. 60.00 ± 10.00 ms, *p* = 0.001). The expression of Cx40 and Cx43 in ventricular tissue was significantly lower in the MS group than in the control group. However, the downregulation of Cx40 and Cx43 was significantly attenuated in the EMPA group compared with the MS and GLI groups. The fibrotic areas of ventricular tissue were also fewer in the EMPA group than that in the MS group. In this study, the ECG QT interval in the EMPA group was shorter than that in the MS group. Compared with the MS group, the EMPA group exhibited significant attenuation of downregulated connexin expression and significantly fewer fibrotic areas in ventricles. These results may provide evidence of possible antiarrhythmic effects of EMPA.

## 1. Introduction

Cardiac arrhythmia is an important risk factor for cardiovascular mortality and morbidity. Previous reports have proven that metabolic syndrome (MS) can modify the electrophysiological characteristics of arrhythmic substrates and be considered a risk factor for cardiac arrhythmia [1,2]. Empagliflozin (EMPA), a sodium–glucose transporter 2 (SGLT2) inhibitor, is a new generation of glucose-lowering agents that inhibit glucose reabsorption in renal tubules [3]. Type 2 diabetes mellitus (DM) patients at high risk for cardiovascular events treated with EMPA had a significantly lower incidence of death from cardiovascular causes and hospitalization for heart failure than patients treated without EMPA in the EMPA-REG trial [4]. Patients with New York Heart Association class II, III, or IV heart failure and an ejection fraction of 40% or less receiving SGLT2 inhibitor therapy also had a significantly lower incidence of death from cardiovascular causes and hospitalization for heart failure than patients without SGLT2 inhibitor therapy in the DAPA-HF trial [5]. According to the consensus report of the American Diabetes Association and the European Association of the Study of Diabetes, SGLT2 inhibitors are considered the first-choice glucose-lowering agent for type 2 DM patients at a high risk for cardiovascular disease and heart failure. The mechanisms of these benefits of SGLT2 inhibitors in patients at a high risk for cardiovascular diseases have not been completely elucidated. In addition to the glucose-lowering effect, SGLT2 inhibitors can reduce the amount of pericardial fat [6], have an antihypertensive effect, and can modify autonomic nervous function [7]. Previous studies have also shown that SGLT2 inhibitors do not prolong the QT interval on electrocardiography, alleviate atrial remodeling, and improve mitochondrial function in diabetic rats [8,9]. We previously demonstrated that EMPA could modulate the effects of adipocytokines from fat tissues on the ion currents of cardiomyocytes and reduce arrhythmogenesis [10]. The effects of SGLT2 inhibitors, especially EMPA, on ventricular electrophysiological substrates have not been elucidated completely. The purpose of this study was to determine and compare the effects of glucose-lowering agents on ventricular electrophysiological substrates in metabolic mice.

## 2. Results

### 2.1. Characteristics of the Metabolic Profiles of the Study Groups

Table 1 demonstrates the characteristics of the metabolic profiles in the study groups after 16 weeks of feeding. Compared with those in the control group, the mean body weights in the metabolic syndrome (MS), empagliflozin (EMPA), and glibenclamide (GLI) groups were significantly increased (29.58 ± 0.46 g vs. 44.62 ± 3.13 g, 37.08 ± 3.30 g, and 44.46 ± 3.05 g, respectively; all *p* < 0.05). The mean body weight in the EMPA group was significantly lower than those in the MS and GLI groups (*p* < 0.05). The mean body weights in the GLI and MS groups were not significantly different. Compared with that in the control group, the mean fasting glucose value in the MS group was significantly increased (164.10 ± 34.35 mg/dL vs. 232.72 ± 48.4 mg/dL, *p* < 0.05), and the mean fasting glucose values were not significantly different in the EMPA and GLI groups. The mean values of cholesterol, low-density lipoprotein (LDL), and high-density lipoprotein (HDL) in the MS, EMPA, and GLI groups were significantly increased compared with those in the control group (all *p* values < 0.05). The HDL level in the MS group was significantly lower than that in the GLI group (45.12 ± 3.59 mg/dL vs. 51.53 ± 2.01 mg/dL, *p* < 0.05) and significantly higher than that in the EMPA group (45.12 ± 3.59 mg/dL vs. 33.63 ± 9.52 mg/dL, *p* < 0.05). The triglyceride levels in the MS and GLI groups were significantly increased compared with those in the control group (20.80 ± 2.95 mg/dL, 19.00 ± 2.31 mg/dL vs. 11.50 ± 1.73 mg/dL, *p* < 0.05). The triglyceride levels were not significantly different between the control and EMPA groups. The different body weight and lipid profiles suggested that the metabolic mice were successfully created by consumption of a high-fat diet for 16 weeks.

### 2.2. Characteristics of the Echocardiography and Electrocardiography (ECG) in the Ventricles of the Study Groups

The characteristics of echocardiography in the ventricles, including right ventricular dimension at end-diastole (RVDd), interventricular septum thickness at end-diastole (IVSd), left ventricular internal diameter at end-diastole (LVIDd), left ventricular posterior wall thickness at end-diastole (LVPWd), interventricular septum thickness at end-systole (IVSs), left ventricular internal diameter at end-systole (LVIDs), left ventricular posterior wall thickness at end-systole (LVPWs), and ejection fraction, were all not significantly different among the study groups after feeding for 16 weeks (Table 2). Figure 1 shows an example of measurement in the left ventricle among the study groups. Table 3 demonstrates the characteristics of ECG and the effective refractory period (ERP) of ventricles. The QRS wave duration and RR interval of ECG were not significantly different among the study groups. The QT interval was significantly shorter in the EMPA group than in the MS group (50.00 ± 2.62 vs. 57.06 ± 3.43 ms; *p* = 0.011) and not significantly different in the control (50.00 ± 2.62 vs. 53.66 ± 2.32 ms; *p* = 0.519) and the GLI (50.00 ± 2.62 vs. 56.87 ± 4.04 ms; *p* = 0.061) groups. The QT intervals among the control, MS and GLI groups were not significantly different after multiple comparisons (all *p* > 0.05). The corrected QT intervals (QTc) among the control, MS, EMPA, and GLI groups were also not significantly different after multiple comparisons. Figure 2 demonstrates an example of measuring characteristics of ECG in the ventricles among the study groups. The ERP in the right ventricle (RV) was not significantly different among the study groups. However, the ERP in the left ventricle (LV) was significantly shorter in the EMPA group (20.0 ± 10.0 ms) than in the GLI (60.0 ± 10.0 ms) after multiple comparisons, respectively (*p* = 0.001). Ventricular arrhythmias could be induced in one mouse in the control, EMPA, and GLI groups, respectively.

### 2.3. Expression of Connexin 40 (Cx40) and Connexin 43 (Cx43) in Cardiac Ventricles

To monitor the effect of EMPA in regulating Cx protein expression in ventricular tissue, histological analyses of immunofluorescence and immunohistochemistry (IHC) staining were conducted. Figure 3 and Figure 4 demonstrate the expression of Cx40 and Cx43 in the ventricles of the study groups. Figure 3A shows the confocal images of Cx40 in LV and RV among the study groups. In LV tissues, the Cx40 expression in the MS group was significantly decreased compared with that in the control group (Figure 3A,B, left panel). The Cx40 expression in the EMPA group was significantly higher than that in the MS group. Likewise, the MS group expressed low levels of Cx40 in the RV tissue, while it expressed relatively high levels of Cx40 in the EMPA group (Figure 3A,B, right panel). There was no difference in Cx40 expression between the control and EMPA groups. However, the GLI treatment group expressed a relatively low level of Cx40 in LV and RV tissue compared with the control and EMPA groups, and there was no significant difference between the MS and GLI groups (Figure 3A,B). Besides, IHC results also support the reduced Cx40 expression in MS group LV and RV tissue compared with the control group, and the relatively high expression of that in the EMPA treatment group (Figure 3C).

Histological analysis of Cx43 expression in ventricular tissue is revealed in Figure 4. The expression of Cx43 in the LV and the RV of the MS group were both decreased. The Cx43 expression level of the EMPA treatment group was higher than that of the MS group, and there was no significant difference compared to the control group (Figure 4A,B). For the same regulation effect of GLI on Cx40, there was no significant difference in Cx43 expression between the GLI group and the MS group. IHC results showed the same effect of EMPA in expressing higher levels of Cx43 expression in ventricular tissue compared with the MS group (Figure 4C). The results indicate that MS caused the downregulation of Cx40 and Cx43 in LV and RV tissues, and EMPA appeared to restore Cx40 and Cx43 to normal expression levels.

### 2.4. Fibrotic Areas of the Study Groups

The fibrotic areas of the study groups were also evaluated. Figure 5A showed longitudinal sections of LV and RV stained with Mason’s trichrome. In LV tissues, the fibrotic areas in the MS were significantly increased compared with that in the control group. The fibrotic areas of the EMPA and GLI groups were significantly fewer than those in the MS group. In RV tissues, the fibrotic areas in the MS, EMPA, and GLI groups were increased compared with those in the control group. The fibrotic areas in the EMPA group were also significantly fewer than those in the MS group. There was no significant difference in the fibrotic areas between the EMPA and GLI groups (Figure 5B).

## 3. Discussion

### 3.1. General Discussion

There are several main findings in our study. First, the QT interval of ECG in the EMPA group was significantly shorter than that in the MS group. The ERP of the LV in the EMPA group was also shorter than that in the GLI group. These changes were found in the absence of significant structural differences in echocardiography. Second, EMPA significantly increased the expression of Cx40 and Cx43 in the LV and RV tissues compared with that of the MS group. Finally, in both LV and RV tissues, the fibrotic areas in the EMPA group were significantly fewer than those in the MS group. There was no significant difference in the fibrotic areas between the EMPA and GLI groups.

### 3.2. The Change of Electrophysiology and Electrocardiography in the Metabolic Syndrome

MS is a well-known risk factor for cardiac arrhythmias. In previous reports, MS could increase the pericardial fat volume, which is associated with an increased incidence of cardiac arrhythmias [11,12]. Adipocytokines of pericardial fat have also been shown to be involved in cardiac arrhythmias. Lee et al. demonstrated that adipocytokines from pericardial fat significantly decreased the delayed-rectifier potassium current (*I*_K_) in ventricular myocytes, which might cause cardiac arrhythmia [13]. At the same time, it also increased the fibrosis area in ventricular tissues in individuals with metabolic syndrome [14]. This structural remodeling could decrease conduction velocity and increase conduction heterogenicity in heart tissue, which could contribute to arrhythmogenesis [15]. For patients with MS, the prolongation of QTc was also observed [16]. Taken together, the evidence above shows that MS could cause structural and electrophysical remodeling to result in arrhythmogenesis. In our study, there was no statistical difference between the control and MS groups in echocardiography, ECG, and induced ventricular arrhythmia. The QT interval could be affected by many factors such as the distribution and amount of gap junction, and the function of intraventricular and interventricular conduction tissues. Alternations in the characteristics of electrophysiology and ECG were the sum effects of these factors. We did not evaluate these factors in the whole ventricular substrate. These could be the mechanisms because of which there was no statistical difference between the control and MS groups in echocardiography, QRS duration, RR interval, QTc interval, ERP, and induced ventricular arrhythmia in this study. 

### 3.3. The Change in Electrophysiology and Electrocardiography after Sulfonylurea Treatment

Sulfonylurea, a traditional glucose-lowering drug, was the most commonly used dual-therapy add-on to metformin in type 2 DM. It was also known to inhibit the adenosine triphosphate-sensitive potassium channels to stimulate insulin secretion. In addition to glucose-lowering effects, it could decrease inflammation and fibrotic areas of cardiomyocyte [17]. Sulfonylurea might attenuate ischemic preconditioning and prevent the shortening of action potential duration, which results in the propagation of delayed afterdepolarizations and prevents re-entrant arrhythmias by inhibiting adenosine triphosphate-sensitive potassium channels [18]. Sulfonylurea also had been elucidated to decrease inflammatory reactions and cardiac fibrosis in streptozotocin-induced diabetic rats [19]. However, some sulfonylureas had extra-pancreatic effects, which include human ether-a-go-go-related gene (hERG) channel inhibition, and resulted in electrocardiographic QT interval prolongation [20,21]. In addition, sulfonylurea might also inhibit *I*_K_ to increase the vulnerability to cardiac arrhythmias [22]. These different properties might either propagate or prevent ventricular arrhythmias. According to Argatroban in the Acute Myocardial Infarction 2 (ARGAMI-2) trial [23], there was no significant difference in in-hospital ventricular tachycardia and/or ventricular fibrillation between sulfonylurea users and the persons treated with diet alone (8.3% vs. 8.9%, *p* = 0.21). There was also no significant difference in sudden death risk of intensive sulfonylurea treatment from the results of the United Kingdom Prospective Diabetes Study (UKPDS) [24]. In our study, the ERP of the LV in the GLI group was statistically longer than the EMPA group. The fibrotic areas in the GLI group were fewer than those in the MS group, which is compatible with previous reports [19]. The Cx40 and Cx43 expressions were significantly lower in the GLI group than in the control and EMPA groups. Some of these effects on the characteristics of ventricular electrophysiological substrates by sulfonylureas might provide mechanisms for the pro-arrhythmic and antiarrhythmic properties of sulfonylureas. The sulfonylurea could not elucidate the benefits of cardiac arrythmia and sudden cardiac death in clinical practice. 

### 3.4. The Change of Electrophysiology and Electrocardiography after EMPA Treatment

SGLT2 inhibitors are new glucose-lowering agents that have been proven to reduce the amount of pericardial fat [6]. These modifications of adipocytokines by SGLT2 inhibitors and sulfonylurea treatment have been elucidated in a previous study [25]. In our previous study, the ion currents of cardiomyocyte could be modulated by adipocytokines after EMPA treatment, which might provide a possible mechanism for the antiarrhythmic effect [10]. In the EMPA-REG study, EMPA also showed benefits for patients with heart failure treatment and for the prevention of sudden cardiac death [4]. However, the impact of EMPA treatment on cardiac electrophysiological characteristics has seldom been discussed. In this study, the modification of electrophysiological characteristics in the EMPA group differed from that in the MS and GLI groups. Compared with the EMPA group, longer QT intervals, lower connexin expression, and larger fibrotic area were found in the MS and GLI groups, and these characteristics could cause cardiac arrhythmia in previous studies [26,27,28]. Compared with those in the MS and GLI groups, the changes in electrophysiological characteristics in the EMPA group were attenuated and more like those in the control group, which might imply that EMPA treatment is beneficial for patients with cardiac arrhythmias.

### 3.5. The Changes in Cx40 and Cx43 Expression in Ventricular Tissue after EMPA Treatment

In previous reports, the association of cardiac arrhythmia and gap junctions was discussed [29,30]. MS reduces the expression of Cx40 and Cx43, which delays the electrical conduction in the atrial and ventricular tissues and might cause cardiac arrhythmias [31]. Cx40 and Cx43 provide low resistance pathways for the electrical activity of cardiomyocytes. In an animal study, the increasing heterogeneity of Cxs in the atrial myocardium was shown to affect the atrial effective refractory period and enhance atrial refractoriness, which resulted in abnormal atrial conduction and arrhythmogenicity [32]. In a previous study, the volume of epicardial fat was shown to be associated with the lateralization of Cx40, which caused aberrant excitability and conduction heterogeneity, resulting in cardiac arrhythmia [33]. In this study, Cx40 and Cx43 were downregulated in the MS group. The downregulation of Cx in metabolic syndrome was found in not only cardiomyocytes [31] but also osteocytes [34]. The downregulation of connexins has also been associated with inflammatory cytokines, such as interleukin-6 (IL-6) and tumor necrosis factor alpha (TNF-α) [35], which might cause cardiac remodeling and fibrosis, resulting in arrhythmogenicity. In this study, the downregulation of Cx expression was attenuated in the EMPA group. Changes in inflammatory cytokines and adipocytokines after EMPA treatment were demonstrated in a previous study [25]. The attenuation of Cx downregulation might result from the modification of inflammatory cytokines after EMPA treatment. In addition, the Cx expression was higher in the EMPA group than in the GLI group, which might imply more antiarrhythmic effects by SGLT2 inhibitors than GLI.

### 3.6. The Changes in Fibrotic Areas of Ventricular Tissue after EMPA Treatment

In our experiment, the cardiac fibrotic effects on the MS and EMPA groups differed. Cardiac fibrosis in metabolic syndrome has been proven by imaging and histological observations in previous reports [31,36]. According to an article published in 2006, these effects might be due to adipose-tissue-resident macrophages of visceral fat, which produce more proinflammatory cytokines [37] and cause cardiac arrhythmia [38,39]. In addition, cardiac fibrosis not only causes cardiac arrhythmia via a trigger, conduction delay, or re-entry circuit [28] but also increases the incidence of heart failure. In this study, the cardiac fibrotic areas of the EMPA were decreased compared with those of the MS group, which implied that EMPA attenuated and possibly protected against cardiac arrhythmia. In previous reports, the sulfonylurea had also been elucidated to decrease cardiac fibrosis [19]. In our experiment, a more significant effect of decreasing fibrosis was found in the EMPA group than in the GLI group, which implied a better effect of SGLT2 inhibitor treatment on heart failure and anti-arrhythmia compared with the sulfonylurea treatment. However, more investigations are needed to further confirm these data.

### 3.7. Limitations

There are several limitations in our study. First, the number of study mice was limited and the study mice may be too young to demonstrate differences in ventricular dimension and thickness. Second, although several characteristics of ventricular electrophysiological substrates after glucose-lowering agent treatment were demonstrated in this study, other electrophysiological characteristics of ventricles, such as the distribution of gap junctions, and electrical conduction properties in the ventricular tissues, were not studied completely, which might result in no statistical difference among study groups including QRS duration, RR interval, QTc interval, ERP and induced ventricular arrhythmia. The alternations of the characteristics of ventricular electrophysiological substrates studied in this study were not sufficient to clarify the mechanisms of ventricular arrhythmogenicity in MS mice with glucose-lowering agent therapy. Third, the serum concentrations of EMPA and GLI in the mice were not determined. Therefore, the therapeutic effects in both the EMPA and GLI groups were unknown. Finally, although the inducibility of ventricular arrhythmias among study groups was evaluated in this study, the results (in Table 3) were not interpreted due to a limited number of study mice in each study group. However, the characteristics of ventricular electrophysiological substrates were determined and compared among the study groups. Our results may only provide indirect evidence regarding the mechanisms of EMPA (or glucose-lowering agents) in the anti-ventricular arrhythmias effect. Whether EMPA has anti-arrhythmias effects needs to be further studied. 

## 4. Materials and Methods

### 4.1. Study Animal Preparation

A total of 20 C57BL/6J mice (age: 8 weeks) were divided into four groups (5 animals/group): (1) control group, mice fed standard chow for 16 weeks; (2) MS group, mice fed a high-fat diet for 16 weeks; (3) EMPA group, mice fed a high-fat diet (Research Diets, New Brunswick, NJ, USA) for 12 weeks and administered EMPA at 10 mg/kg daily for 4 weeks; and (4) GLI group, mice fed a high-fat diet for 12 weeks and administered GLI at 0.6 mg/kg daily for the following 4 weeks. After 16 weeks of feeding, echocardiography, ECG, and biochemistry data as well as the characteristics of ventricular electrophysiological substrates were analyzed and compared among the groups. Venous blood (10 c.c.) was withdrawn from the inferior vena cava to determine the sugar, total cholesterol, triglyceride, LDL, and HDL levels. The study protocol was approved by the Institutional Animal Care and Use Committee at Kaohsiung Medical University (approval IACUC code: KMU-107238, execution data: 2019/8/1-2022/7/31).

### 4.2. Characteristics of ECG and Echocardiography in the Ventricles

For recoding and analyzing the ECG and echocardiography data on the ventricles, each mouse was anesthetized with 1.5% isoflurane inhalation after 16 weeks of feeding. Platinum electrodes were inserted into the limbs subcutaneously and connected to a bioamplifier (IX-TA-220 Recorder, iWorx, Dover, NH, USA), which recorded and stored a single lead ECG signal for 5 min at 200 kHz. For analysis of electrocardiography, the QRS wave duration, RR interval, QT interval, and QTc interval were measured using LabScribe v4 software (iWorx, Dover, NH, USA) on a computer screen at a screen rate of 500 per second. The QTc was calculated by Bazett’s formula, QTc = QT/(RR)^1/2^. Each measurement was performed for three consecutive sinus heartbeats, and the data are presented as the mean value of the three measurements. Echocardiography was performed with a Vevo 2100 parasternal long axis view (VisualSonics, Inc., Toronto, OT, Canada), and the RVDd, IVSd, LVIDd, LVPWd, IVSs, LVIDs, LVPWs, and ejection fraction were analyzed using a Vevo 2100 imaging system (VisualSonics, Inc., Toronto, OT, Canada).

### 4.3. Determination of the ERP of the RV and LV by Langendorff Heart Perfusion

After collection of the ECG and echocardiography measurements, the heart was harvested for measurement of the ERPs of the RV and the LV by the Langendorff heart perfusion method, and 10 c.c. of venous blood was withdrawn from the inferior vena cava to determine the metabolic profiles, including the sugar, total cholesterol, LDL, HDL, and triglyceride levels. The Langendorff heart perfusion protocol has been described in detail in a previous study. In brief, the heart was harvested through a thoracotomy in ice-cold Tyrode’s solution (in mmol/L: NaCl 125, KCl 4.5, NaHCO_3_ 24, NaH_2_PO_4_ 1.8, CaCl_2_ 1.5, MgCl_2_ 0.5, and glucose 5.5) and cannulated via the aorta. The heart was then connected to a Langendorff apparatus and perfused with Tyrode’s solution with 95% O_2_ and 5% CO_2_ to maintain a pH of 7.35~7.45 at 37 °C at a constant pressure of 70–80 mmHg (flow rate: 1–3 mL/min). The pseudo-ECG was performed by using widely spaced bipolar electrodes surrounding the mouse heart in the bath to determine the heart rhythm. To determine the ERPs of the ventricles, a bipolar pacing lead was placed at the epicardium of the RV and LV close to the apex areas.

Stimuli were delivered as rectangular 2 ms pulses at twice the diastolic threshold. The ERPs of the RV and LV were measured by introducing S2 extra-stimuli after eight regularly paced S1–S1 intervals at a basic pacing cycle length of 100 ms. S2 was delivered from 180 ms of the S1–S2 coupling interval at decremental intervals of 10 ms until the ERP was reached. The ERP was defined as the longest S1–S2 interval at which point S2 could not capture the ventricle. The vulnerability of ventricular tachyarrhythmia was defined as the maximal duration of repetitive ventricular responses induced by the S2 extra-stimulus during the measurement of the ventricle ERPs or by decremental ventricular pacing with a pacing cycle length from 100 ms to the cycle length of no more capture.

### 4.4. Ventricle Expression of Connexin 40 and Connexin 43

After determining the ERPs of both ventricles, RV and LV tissues were obtained to determine the expression of Cx40 and Cx43, respectively. The expression of Cx40 and Cx43 was determined by immunofluorescence imaging and IHC staining. The protocols used for the immunofluorescence imaging have been described in a previous study [31]. In brief, ventricular tissues were cut and then placed in formalin for paraffin embedding. The paraffin-embedded tissues were sectioned onto slides, dewaxed with 100% xylene, and rehydrated in graded alcohol solutions. Throughout the procedure, the slides were washed in phosphate-buffered saline (PBS). Serial sections were blocked with 10% fetal bovine saline (FBS) in PBS and incubated with the primary monoclonal antibody goat anti-Cx40 (1:50, Santa Cruz Biotechnology Inc., Dallas, TX, USA) or rabbit anti-Cx43 (1:100, Cell Signaling Technology, MA, USA) overnight. The slides were washed, and Wheat Germ Agglutinin Texas Red^®^-X Conjugate (WGA, 1:500, Thermo Fisher Scientific Inc., Waltham, MA, USA) was applied for 1 h at 37 °C for plasma membrane staining. The slides were then incubated with a CF 488A-labeled secondary antibody (1:500; Sigma-Aldrich Inc., St. Louis, MO, USA) for 1 h at room temperature in the dark. Nuclei were counterstained with DAPI (1:10,000, Biotium Inc., San Francisco, CA, USA) before coverslip mounting. Immunofluorescence images were obtained and analyzed on an LSM 700 confocal microscope (Zeiss, Oberkochen, Germany). Images were collected using the ×100 objective lens and Zoom 1.5 computer setting. The images were acquired with 1024 × 1024 resolutions in at least four different regions of each slide analyzed. The fluorescence intensity of the image was calculated using Image-Pro Plus Version 6.0 (Media Cybernetics, Inc., Bethesda, MD, USA). 

### 4.5. Histological and Immunohistochemistry (IHC) Staining

The ventricle sections were harvested, fixed in 4% PFA for 3 days, demineralized with 0.5 M EDTA, and embedded in paraffin. Sections 5 µm thick were sliced and then rehydrated for histological staining as follows: hematoxylin and eosin staining, hematoxylin (MHS1, Sigma-Aldrich, St. Louis, MO, USA) for 120 s, a ddH_2_O wash, and eosin (230251, Sigma-Aldrich) for 3 s; and Safranin O/fast green staining, hematoxylin (MHS1, Sigma-Aldrich) for 180 s, a ddH_2_O wash, 0.05% fast green (2353-45-9, Sigma-Aldrich) for 90 s, and 0.1% Safranin O red (HT90432, Sigma-Aldrich). For IHC staining, antigen retrieval was conducted by boiling the sections in 0.5% Tris-EDTA in 1× PBS for 30 min. Next, the sections were incubated with 3% H_2_O_2_ for 10 min at room temperature, blocked with blocking buffer (ab126587, Abcam, Cambridge, MA, USA) and incubated overnight at 4 °C with primary antibodies targeting Cx40 (Santa Cruz Biotechnology Inc., Dallas, TX, USA) or rabbit anti-Cx43 (Cell Signaling Technology, Danvers, MA, USA). The sections were then treated with mouse- and rabbit-specific HRP/DAB detection kits (ab64264, Abcam, Cambridge, MA, USA), counterstained with hematoxylin and observed under a Leica-DM1750 microscope (Leica Microsystems, Wetzlar, Germany).

### 4.6. The Measurement of Fibrotic Area in Ventricles

The ventricles were sectioned longitudinally to measure the fibrotic areas by Mason’s trichrome staining [40]. Under the microscope, the fibrotic areas appeared green, and muscle areas appeared red. The degree of fibrosis in RV and LV tissues was expressed as the ratio of fibrotic areas to muscle areas as calculated using computer software (Image-Pro Plus, Media Cybernetics, Inc., Bethesda, MD, USA).

### 4.7. Statistical Analysis

All data are expressed as the mean ± standard deviation. The Kruskal–Wallis test was used to compare the characteristics of the study groups. The multiple comparisons with significance were further verified by using Tukey’s test (Studentized range distribution) of post hoc test to avoid type I error. A *p* value < 0.05 was considered significant. All statistical analyses were performed using SPSS software 17.0 (SPSS Inc., Chicago, IL, USA).

## 5. Conclusions

Compared with the MS group, the EMPA group exhibited a shorter QT interval. In addition, higher Cx expression and fewer cardiac fibrotic areas in ventricle tissues were observed in the EMPA group compared with the MS group. Compared with the GLI group, higher Cx expression in the EMPA group was also elucidated. These findings might imply lower arrhythmogenicity in the EMPA group. However, more investigations are needed to further elucidate the possible mechanisms and antiarrhythmic effects of EMPA.

## Figures and Tables

**Figure 1 ijms-22-06105-f001:**
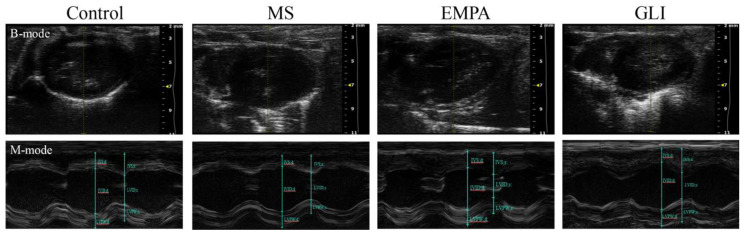
B-mode and M-mode images for the measurement of the dimensions of interventricular septal thickness at end-diastole (IVSd), left ventricular internal diameter at end-diastole (LVIDd), left ventricular posterior wall thickness at end-diastole (LVPWd), interventricular septal thickness at end-systole (IVSs), left ventricular diameter at end-systole (LVIDs), and left ventricular posterior wall thickness at end-systole (LVPWs) among the study groups (*n* = 5 for each group).

**Figure 2 ijms-22-06105-f002:**
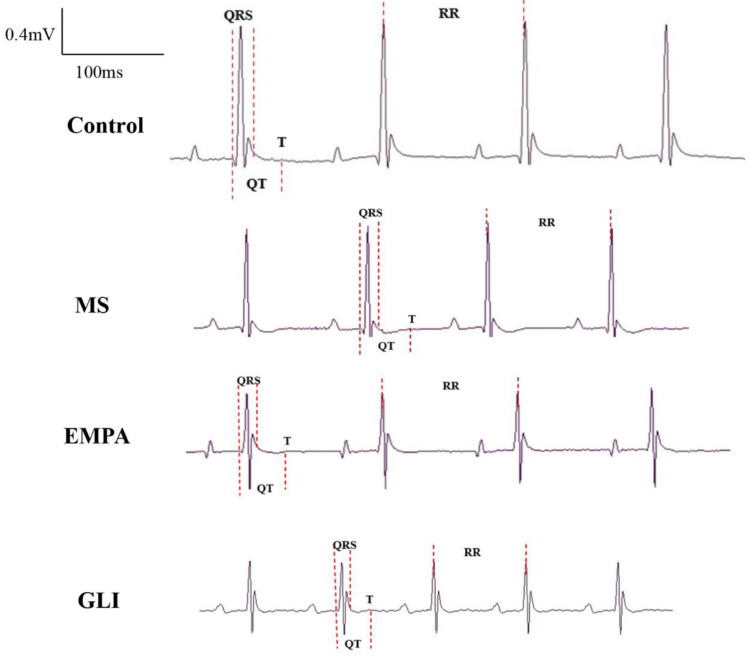
An example of measuring characteristics of electrocardiography (ECG) in the ventricles among the study groups (*n* = 5 for each group). The dashed lines indicate measurements of the QRS wave duration, RR interval, and QT interval.

**Figure 3 ijms-22-06105-f003:**
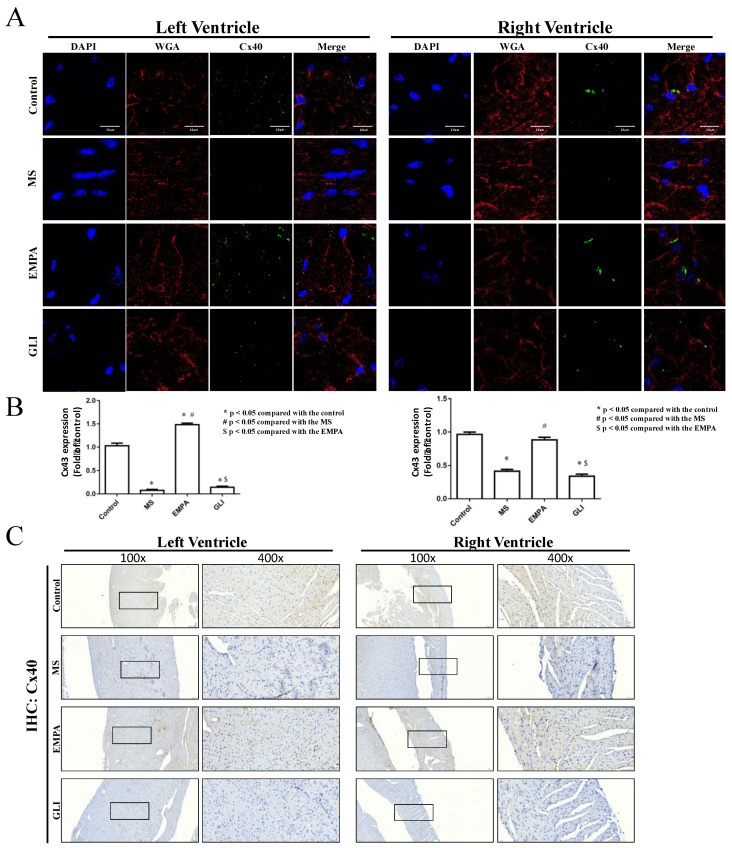
Histological analysis of Cx40 in mouse ventricular tissue. (**A**) Confocal image of ventricles Cx40 immunofluorescence stain in the left ventricle (LV) (left panel) and the right ventricle (RV) (right panel) among the study groups (*n* = 5 for each group). Blue for DAPI (cell nucleus) staining, red for WGA (cell membrane) staining, and green for Cx40-positive staining. The scale bars indicate 10 μm. (Magnification: 1000×.) (**B**) The quantification of Cx40 expression integrated from a series of confocal image sections in the ventricles among the study groups. (**C**) IHC staining of Cx40 in mouse ventricles of each group. Blue for cell nucleus stain and brown for Cx40-positive stain. (Magnification: 100× and 400×.) *: *p* < 0.05 compared with the control group; #: *p* < 0.05 compared with the MS group; $: *p* < 0.05 compared with the EMPA group.

**Figure 4 ijms-22-06105-f004:**
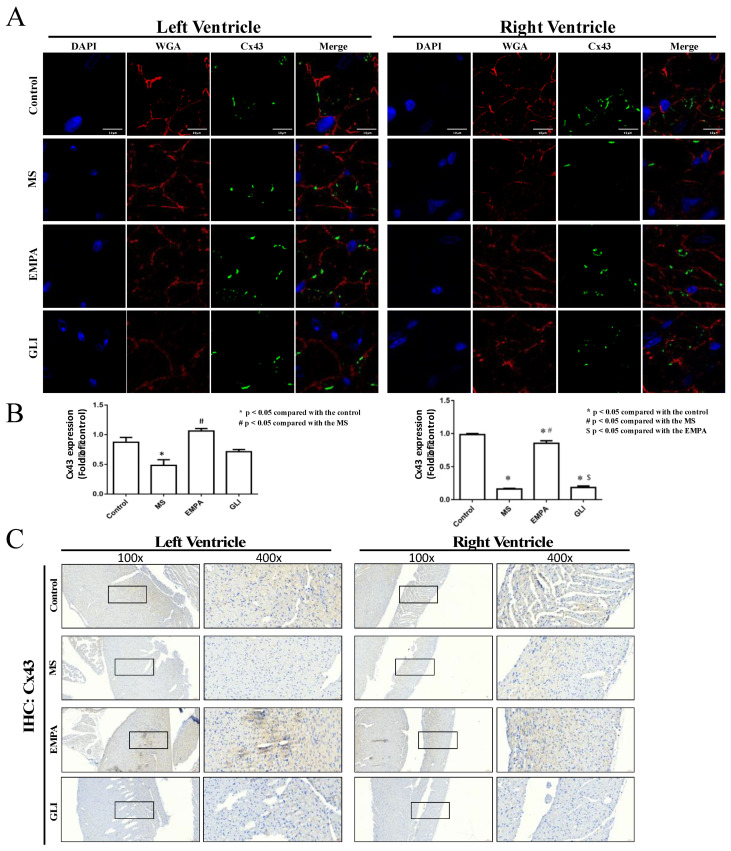
Histological analysis of Cx43 expression in mouse ventricles. (**A**) Immunofluorescence of Cx43 confocal image in the LV (left panel) and the RV (right panel) among the study groups (*n* = 5 for each group). Nuclei with DAPI staining appear blue. The plasma membrane with WGA staining appears red, and green for Cx43-positive staining. The scale bars indicate 10 μm. (**B**)The quantification of Cx43 signaling integrated from a series of confocal image sections in ventricles among the study groups. (**C**) IHC staining of Cx43 in mouse ventricles of each group. Blue is used for cell nucleus staining and brown for Cx43-positive staining. *: *p* < 0.05 compared with the control group; #: *p* < 0.05 compared with the MS group; $: *p* < 0.05 compared with the EMPA group.

**Figure 5 ijms-22-06105-f005:**
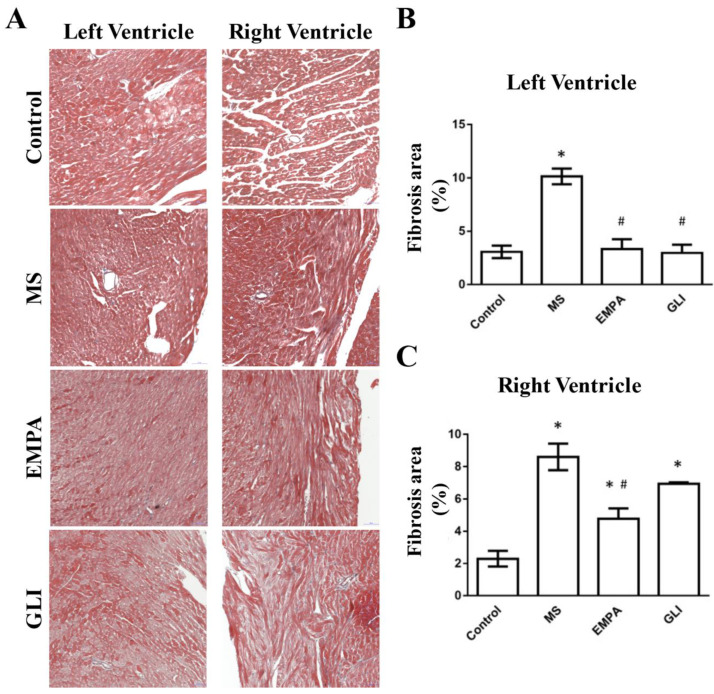
(**A**) Histological fibrotic images of ventricles among the study groups (*n* = 5 for each group) as determined by Mason’s trichrome staining. Bar graphs of fibrotic areas in ventricles. (**B**) In the LV, the fibrotic areas in the MS were significantly increased compared with that in the control group. The fibrotic areas of the EMPA and GLI groups were significantly fewer than those in the MS group. **(C**) In the RV, the fibrotic areas in the MS, EMPA, and GLI groups were increased compared with those in the control group. The fibrotic areas in the EMPA group were still significantly fewer than those in the MS group. *: *p* < 0.05 compared with the control group; #: *p* < 0.05 compared with the MS group.

**Table 1 ijms-22-06105-t001:** Characteristics of the study groups after 16 weeks of feeding.

	Control	MS	EMPA	GLI
Weight (g)	29.58 ± 0.46	44.62 ± 3.13 *	37.08 ± 3.30 *^,#^	44.46 ± 3.05 *^,†^
Fasting glucose (mg/dL)	164.10 ± 34.35	232.72 ± 48.4 *	200.23 ± 13.74	211.62 ± 28.66
Cholesterol (mg/dL)	47.5 ± 2.65	115.60 ± 13.09 *	99.75 ± 15.69 *	133.00 ± 4.24 *^,#,†^
LDL (mg/dL)	2.55 ± 0.17	5.35 ± 0.29 *	5.12 ± 0.95 *	7.15 ± 1.84 *
HDL (mg/dL)	20.45 ± 2.37	45.12 ± 3.59 *	33.63 ± 9.52 *^,#^	51.53 ± 2.01 *^,#,†^
Triglyceride (mg/dL)	11.50 ± 1.73	20.80 ± 2.95 *	17.25 ± 8.02	19.00 ± 2.31 *

MS: metabolic syndrome group; EMPA: metabolic syndrome with empagliflozin therapy group; GLI: metabolic syndrome with glibenclamide therapy group; HDL: high-density lipoprotein; LDL: low-density lipoprotein; *: *p* value < 0.05 compared with the control group; ^#^: *p* value < 0.05 compared with the MS group; ^†^: *p* value < 0.05 compared with the EMPA group.

**Table 2 ijms-22-06105-t002:** Characteristics of echocardiography in the study groups.

	Control	MS	EMPA	GLI
RVDd (mm)	1.08 ± 0.26	1.03 ± 0.08	0.94 ± 0.14	0.96 ± 0.16
IVSd (mm)	0.67 ± 0.06	0.72 ± 0.06	0.74 ± 0.08	0.66 ± 0.03
LVIDd (mm)	2.81 ± 0.28	2.85 ± 0.09	2.73 ± 0.21	2.76 ± 0.12
LVPWd (mm)	0.79 ± 0.02	0.85 ± 0.16	0.81 ± 0.06	0.79 ± 0.09
IVSs (mm)	0.84 ± 0.09	0.96 ± 0.09	0.90 ± 0.06	0.98 ± 0.11
LVIDs (mm)	1.81 ± 0.22	1.67 ± 0.30	1.63 ± 0.17	1.75 ± 0.34
LVPWs (mm)	0.84 ± 0.13	1.00 ± 0.17	0.98 ± 0.07	0.90 ± 0.14
Ejection fraction (%)	58.39 ± 6.93	64.71 ± 12.94	64.31 ± 2.97	59.23 ± 13.06

RVDd: right ventricular diameter at end-diastole; IVSd: interventricular septal thickness at end-diastole; LVIDd: left ventricular internal diameter at end-diastole; LVPWd: left ventricular posterior wall thickness at end-diastole; IVSs: interventricular septal thickness at end-systole; LVIDs: left ventricular internal diameter at end-systole; LVPWs: left ventricular posterior wall thickness at end-systole.

**Table 3 ijms-22-06105-t003:** ECG parameters and effective refractory periods (ERPs) of the study groups.

	Control	MS	EMPA	GLI
QRS wave duration (ms)	24.27 ± 2.04	23.73 ± 2.55	22.13 ± 2.24	23.33 ± 3.83
RR interval (ms)	144.06 ± 14.87	146.67 ± 21.05	140.40 ± 23.18	142.73 ± 17.91
QT interval (ms)	53.66 ± 2.32	57.06 ± 3.43	50.00 ± 2.62 ^#^	56.87 ± 4.04
QTc interval (ms)	142.29 ± 9.58	149.47 ± 8.25	134.15 ± 6.72	150.86 ± 9.47
ERP of right ventricle (ms)	35.00 ± 10.00	40.00 ± 14.14	47.50 ± 5.00	48.00 ± 13.04
ERP of left ventricle (ms)	35.00 ± 10.00	38.00 ± 4.47	20.00 ± 10.00	60.00 ± 10.00 ^†^
Induced VT/VF mice	1	0	1	1

QTc: corrected QT interval; ERP: effective refractory period; VT: ventricular tachycardia; VF: ventricular fibrillation; ^#^: *p* < 0.05 compared with the MS group; ^†^: *p* < 0.05 compared with the EMPA group.

## Data Availability

Not applicable.

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
