# Peer review of "Characteristics of Ventricular Electrophysiological Substrates in Metabolic Mice Treated with Empagliflozin"

_ijms, 2021, doi:10.3390/ijms22116105_

Round 1

Reviewer 1 Report

The study by Jhuo et al. considered an important problem of arrhythmogenesis in metabolic syndrome and in conditions of metabolic syndrome treatment. The authors demonstrated that empagliflozin alleviated connexin downregulation and myocardial fibrosis that might result in the amelioration of the myocardial arrhythmogenic substrate. In addition, ERP and QTc shortening by empagliflozin may contribute to the change of arrhythmogenic properties. The background of the study is justified and interesting results are presented. However, there are some issues that should be elaborated on. 

  1. My major concern is that arrhythmogenicity could not be evaluated in this model. Just one or no at all VT/VF episode that could be evoked indicates that the experimental conditions were not arrhythmogenic enough to draw any conclusions about the arrhythmogenic substrate.
  2. Four studied groups, five animals each, provide quite a limited statistical power, and only rather drastic differences can be found in such a design. Moreover, the authors wrote (lines 366-367): "Continuous variables in the two groups were compared using the nonparametric independent two-sample test (Mann-Whitney U test)." It looks like that no multiple comparison corrections were applied. My guess is if correct multiple comparisons are done in the set of 4*5 animal groups, probably no differences would have been found. 
  3. The authors showed that empagliflozin ameliorates connexin downregulation and fibrosis. If this observation is correct (see the comment above), it is expected that these effects should have had some functional expression, e.g. in QRS duration or conduction velocity. The conduction velocity was not tested and QRS duration did not differ between the groups. This discrepancy should at least be commented on. 
  4. It should be taken into account that ERP and QT shortening that has been demonstrated in the study might have both arrhythmic and antiarrhythmic consequences. 

Author Response

Response to Reviewers

Reviewer 1

The study by Jhuo et al. considered an important problem of arrhythmogenesis in metabolic syndrome and in conditions of metabolic syndrome treatment. The authors demonstrated that empagliflozin alleviated connexin downregulation and myocardial fibrosis that might result in the amelioration of the myocardial arrhythmogenic substrate. In addition, ERP and QTc shortening by empagliflozin may contribute to the change of arrhythmogenic properties. The background of the study is justified, and interesting results are presented. However, there are some issues that should be elaborated on. 

  1. My major concern is that arrhythmogenicity could not be evaluated in this model. Just one or no at all VT/VF episode that could be evoked indicates that the experimental conditions were not arrhythmogenic enough to draw any conclusions about the arrhythmogenic substrate.

Response:

We appreciate your valuable opinion. Due to limited number of study mice, effects of ventricular arrhythmogenicity by glucose-lowering agents were not directly evaluated in detail in this study. However, effects of glucose-lowering agents on the electrophysiological ventricular substrates including the QT interval, ventricular fibrosis areas, amount of connexin in ventricles were demonstrated in this study. Together with these results, empagliflozin could reduce ventricular arrhythmogenic substrates in metabolic mice and show the potential of empagliflozin in reducing ventricular arrhythmias. The related description had been added to the Discussion section of main text.
(Discussion section, line 208-215)

  1. Four studied groups, five animals each, provide quite a limited statistical power, and only rather drastic differences can be found in such a design. Moreover, the authors wrote (lines 366-367): "Continuous variables in the two groups were compared using the nonparametric independent two-sample test (Mann-Whitney U test)." It looks like that no multiple comparison corrections were applied. My guess is if correct multiple comparisons are done in the set of 4*5 animal groups, probably no differences would have been found. 

Response:

We thank for your valuable opinions. Limited number of study animals is a limitation in our study. The Kruskal-Wallis test was used to correct multiple comparison of the study groups. After multiple comparison of our variables, the QT interval in EMPA group was still significantly shorter than in the MS groups (50.00±2.62 vs. 57.06±3.43 ms; p=0.011). The ERP of LV in the GLI group was significantly longer than that in the EMPA group. It might reveal different characteristics of substrate after EMPA treatment. The table 3 had been added and statistical method had been revised in Material and methods section.

(Material and methods section 4.7, line 395-397.)

Table 3. ECG Parameters and Effective Refractory Periods (ERPs) of the Study Groups

Control

MS

EMPA

GLI

QRS wave duration (ms)

24.27±2.04

23.73±2.55

22.13±2.24

23.33±3.83

RR interval (ms)

144.06±14.87

146.67±21.05

140.40±23.18

142.73±17.91

QT interval (ms)

53.66±2.32

57.06±3.43

50.00±2.62#

56.87±4.04

QTc interval (ms)

142.29±9.58

149.47±8.25

134.15±6.72

150.86±9.47

ERP of right ventricle (ms)

35.00±10.00

40.00±14.14

47.50±5.00

48.00±13.04

ERP of left ventricle (ms)

35.00±10.00

38.00±4.47

20.00±10.00

60.00±10.00†

Induced VT/VF mice

1

0

1

1

QTc: corrected QT interval; ERP: effective refractory period; VT: ventricular tachycardia; VF: ventricular fibrillation; *: p<0.05 compared with the control group; #: p<0.05 compared with the MS group; †: p<0.05 compared with the EMPA group

  1. The authors showed that empagliflozin ameliorates connexin downregulation and fibrosis. If this observation is correct (see the comment above), it is expected that these effects should have had some functional expression, e.g. in QRS duration or conduction velocity. The conduction velocity was not tested and QRS duration did not differ between the groups. This discrepancy should at least be commented on. 

Response:

We thank for your opinion very much. The QRS duration represented the duration of whole ventricular depolarization and the QT and QTc intervals represented duration of ventricular depolarization and repolarization. Many factors could affect the QRS duration, QT and QTc intervals including amount and distribution of gap junction, function of interventricular and intraventricular conduction tissues, and amount and distribution of ventricular fibrosis areas. Alternations of QRS duration, the QT and QTc intervals were sum effects of these factors. Intra- and interventricular conduction velocity, amount and distribution of gap junction, and fibrosis areas in whole ventricular tissues were not evaluated in this study. The related discussion had been revised in the limitation section, which is worthy of further study in the future.

(Discussion section, line 208-215 and line 291-399.)

  1. It should be taken into account that ERP and QT shortening that has been demonstrated in the study might have both arrhythmic and antiarrhythmic consequences. 

Response:

We thank for your valuable opinion. Prolongation of the QT interval could increase vulnerability of ventricular arrhythmias, and a shorter QT intervals were found in the EMPA group than that in the MS group in this study. Besides, by through multiple comparison, there was no significant difference of LV and RV ERP between EMPA and MS group. However, ventricular substrate is not the only evaluation criterion for ventricular arrhythmia. Multiple factors may also affect ventricular arrhythmia, including conduction velocity, distribution of gap junction and fibrosis areas in whole ventricular tissues, and those features still remain to be evaluated in the future. The related description and discussion had been added in the Limitation section.
(Discussion section, line 291-399.)

Reviewer 2 Report

I think the manuscrit is well structured and clearly written, however my major concerns are about the results from section 2.3 and how the quantifications are made in IF Cx40 and Cx43.

  1. Figure 3 showed represententavie images of Cx40 IF along the 4 groups. I am concerned about the images selected. They did not reflect the difference between groups that they claim in the quantification. For this reason would suggest choose better representative images in figure 3 or higher quality images, if that it the problem.
  2. Higher magnification images would be needed
  3. Ideally, I woul like to see Cx40 and Cx43 labeling also by IHC, if possible.
  4. I definite think that is necessary to add more detailed information in section 4.4 of methods about how the quantification of images were perfomed

I really think it is important to clarify this results before can be considered to be published.

Author Response

Response to Reviewers

Reviewer 2

I think the manuscript is well structured and clearly written, however my major concerns are about the results from section 2.3 and how the quantifications are made in IF Cx40 and Cx43.

  1. Figure 3 showed representative images of Cx40 IF along the 4 groups. I am concerned about the images selected. They did not reflect the difference between groups that they claim in the quantification. For this reason, would suggest choosing better representative images in figure 3 or higher quality images, if that is the problem.

Response:

Thanks for the valuable comments. The fluorescent image that best represents the quantitative results has been selected, and the resolution of the image has also been improved to make the results clear and easier to interpret.

(Figure 3A and Figure 4A)

Figure 3A:

Figure 4A:

  1. Higher magnification images would be needed

Response:

The higher magnification images (×100 objective lens and zoom 1.5 computer setting) has been provided to make our results easier to observe. The related microscope analysis conditions were also modified in the Materials and Methods section.

(Materials and Methods section 4.4, line 371-374)

  1. Ideally, I would like to see Cx40 and Cx43 labeling also by IHC, if possible.

Response:

We also agree that IHC is an experiment to better investigate the Cx protein regulation in whole ventricular tissue. The results were also integrated in Figures 3C and 4C, which showed the same results as confocal fluorescent staining. Cx40 and Cx43 were decreased in the MS group, while this two Cx expression in EMPA group was significantly improved compared with MS group, and there was no difference between EMPA and control group. The result further supports our hypothesis, indicating that EMPA can effectively attenuate the decreased expression of Cx40 and Cx43 caused by MS. Related description of results and methods have been revised in the Results, Figure legend, and Materials and Methods section.

(Results 2.3, line 126-139, 140-148; Figure legend, line 151-159 and 162-168; Materials and Methods section 4.5, line 375-388)

Figure 3C:

Figure 4C:

  1. I definite think that is necessary to add more detailed information in section 4.4 of methods about how the quantification of images was performed

Response:

The software and methods for quantification of fluorescent images have been added in section 4.4. After the 1024x1024 image is captured, The Image-Pro Plus version 6.0 was used to analyze the fluorescence intensity of the image and output the data to make histograms and statistical data.
(Materials and Methods section 4.4, line 371-374)

Round 2

Reviewer 1 Report

The efforts of the authors are acknowledged. However, my major concern is still unmet. It is if there were no arrhythmias, it is impossible to say anything about the arrhythmogenic substrate.

Author Response

Response to Reviewer

Reviewer 1

The efforts of the authors are acknowledged. However, my major concern is still unmet. It is if there were no arrhythmias, it is impossible to say anything about the arrhythmogenic substrate.

Response:

Thanks for reviewer’s comments. In clinical practice, MS is a risk factor for arrhythmia, but not all patients with MS will develop arrhythmia. MS is not the only clear cause of arrhythmia. Therefore, the influence of other factors on ventricular substrates and their correlation with arrhythmia should be considered.
       The purposes of this study were to characterize the ventricular electrophysiological substrates in metabolic mice and the effects of glucose-lowering agent. The title of the current manuscript has been revised to "Characteristics of ventricular electrophysiological substrates in metabolic mice treated with empagliflozin ", hoping to be closer to the results and interpretation presenting in our manuscript. Our results may only provide indirect evidence on the mechanism of EMPA (or glucose-lowering agents) against ventricular arrhythmia.
       Although inducibility of ventricular arrhythmias among different study groups was ever evaluated in this study, results (in table 3) were not interpreted due to limited number of study mice in each study group. However, characteristics of ventricular electrical substrates, including QRS duration, QT interval, QTc interval, ERP of ventricles, degree of fibrosis area and expressions of connexins in the ventricles, were determined and compared among study groups.
       Whether EMPA (or glucose-lowering agent) have anti-arrhythmias effects need to more further studies. We have revised the manuscript in page 14, line 302 as “Finally, although inducibility of ventricular arrhythmias among study groups………………….”.